# A Unified Test for the AR Error Structure of an Autoregressive Model

**Xinyi Wei** [1,2], **Xiaohui Liu** [1,2], **Yawen Fan** [1,2,*] , **Li Tan** [1,2] and **Qing Liu** [1,2]

1   School of Statistics, Jiangxi University of Finance and Economics, Nanchang 330013, China
2   Key Laboratory of Data Science in Finance and Economics, Jiangxi University of Finance and Economics, Nanchang 330013, China
*   Correspondence: 2202010084@stu.jxufe.edu.cn

**Abstract:** A direct application of autoregressive (AR) models with independent and identically distributed (iid) errors is sometimes inadequate to fit the time series data well. A natural alternative is further to assume the model errors following an AR process, whose structure however has essential impacts on the statistical inferences related to the autoregressive models. In this paper, we construct a new unified test for checking the AR error structure based on the empirical likelihood method. The proposed test is desirable because its limit distribution is always chi-squared regardless of whether the autoregressive model is stationary or non-stationary, with or without an intercept term. Some simulations are also provided to illustrate the finite sample performance of this test. Finally, we apply the proposed test to a financial real data set.

**Keywords:** autoregressive model; AR errors; empirical likelihood; unified test

**MSC:** 62M10; 91B84; 01-08; 65C60





## 1. Introduction

When auxiliary variables are not available, autoregressive models are widely used to model this kind of time series data. Typically, the response is often assumed to depend linearly on its previous values. Among all autoregressive models, the autoregressive model of the first order, i.e., AR(1), is the simplest, which takes the following form:

$$X_t = \mu + \phi X_{t-1} + \varepsilon_t, \quad t = 1, 2, \cdots, n, \tag{1}$$

where $\mu$ and $\phi$ are unknown parameters with $\mu$ being the intercept item and $\phi$ the autoregression coefficient, and $\{\varepsilon_t\}$ denotes the sequence of random errors or innovations having means of zero.

In many previous studies, a considerable amount of work has been provided on statistical inferences [1–6] and related applications [7–9] for AR models. In terms of practical applications, AR models are commonly used to describe the behavior of inflation or logarithmic exchange rate, where people are interested in whether there is a unit root or persistence of related variables. However, a precondition for an accurate unit root test or persistence test is that the model is properly fitted so that the parameters can be reasonably estimated. To guarantee this, it is important to perform predefined tests, e.g., the unit root test and serial correlation test, on the rationality of using the AR model before conducting a relevant economic analysis.

Among them, the unit root test is the most commonly mentioned. Note that the limit distributions of the estimators of $\mu$ and $\phi$ depend on whether the process $\{X_t\}$ is stationary or non-stationary, i.e., Case (i) $|\phi| < 1$ (stationary), Case (ii) $\mu = 0$ and $\phi = 1 + \frac{c}{n}$ for some nonzero constant $c$ (nearly integrated if $c \neq 0$, and unit root if $c = 0$), and Case (iii) $\mu \neq 0$ and $\phi = 1 + \frac{c}{n}$ for some nonzero constant $c$ (nearly integrated if $c \neq 0$). It is well

known that when the AR process has a unit root, its many statistical procedures have quite complex limit distributions, differing from that for the stationary case. Hence, various testing methods have been developed in the past decades to address the issue of unit root, including the augmented Dickey–Fuller (ADF) test [10], the Phillips–Perron (PP) test [11], the DF–GLS test [3], and the KPSS test [12], etc.

It is worth mentioning that if the true underlying innovations are correlated, the finite sample performance of the tests above may be greatly affected. To improve the efficiency of the estimation, a natural idea is to take into account the special structure of the errors if available. Note that it is common to assume that the errors further follow an AR process when they are correlated, while the performance of some testing procedures can be greatly improved once the AR structure has been addressed sufficiently, as shown in [13,14].

In detail, Ref. [13] considered the following autoregressive model with AR errors:

$$\begin{cases} X_t = \mu + \phi X_{t-1} + \varepsilon_t, \\ e_t = \varepsilon_t + \sum_{i=1}^{p} \psi_i \varepsilon_{t-i}. \end{cases} \tag{2}$$

where $\boldsymbol{\psi} = (\psi_1, \psi_2, \cdots, \psi_p)^\top$ denotes the vector of unknown parameters involved in the AR errors, and $e_t$ denotes the random error involved in $\varepsilon_t$. Compared to Model (1), Ref. [13] here further assumed that $\varepsilon_t$ follows an AR process. Note that (2) implies $\varepsilon_t = \sum_{i=1}^{p} (-\psi_i)\varepsilon_{t-i} + e_t$. A unified unit root test was developed by considering the special structure in $\{e_t\}$. Their test has been shown to have desirable properties, as the related statistic converges in distribution to a standard chi-squared distributed variable. However, their test depends on preconditions such that the AR structure of $\{\varepsilon_t\}$ has been well specified, and $p$ is properly predefined. The violation of these conditions may result in power loss in this method, as shown in our simulations.

To this end, we are interested in producing statistics to test whether $\boldsymbol{\psi}$ is equal to some given constant vector $\boldsymbol{\psi}_0$ under Cases (i)–(iii), which has not been considered in the literature to the best of our knowledge. Note that although many tests have been developed for testing the possible serial correlation in $\{\varepsilon_t\}$, including the Lagrangian multiplier (LM) test [15], Box–Pierce (BP) test [16], and Ljung–Box (LB) test [17], etc., they cannot be used directly to test the hypothesis above. In view of this, we propose an empirical likelihood-based statistic for testing this issue by taking into account the AR structure. Note that the setting in Case (ii) causes issues in the derivation of the asymptotic distribution, as well as the related applications. A new data-splitting idea is also employed in order to unify Cases (i)–(iii). It turns out that the proposed statistic converges in distribution to a standard chi-squared distributed variable regardless of $\{X_t\}$ being stationary or non-stationary, due to the special block structure of the asymptotic covariance matrix. The simulations show that our method has a good size, as well as nontrivial power performance in finite sample cases.

As a nonparametric method, empirical likelihood (EL) was firstly proposed by [18]. Because of its many excellent properties, i.e., no need to assume the parameter distribution in advance, it has been widely used in the literature when parametric methods do not work well to produce satisfactory results. Many authors have devoted themselves to extending this method. To name but a few, Ref. [19] obtained confidence regions for vector-valued statistical functions, which is a multivariate generalization of the work of [18]. Refs. [20,21] extended the empirical likelihood method to the setting of regression models and general estimation equations, respectively. Recently, Ref. [22] discussed the possibility of constructing unified tests by using empirical likelihood based on a weighted technique for time series models. Ref. [13] further extended this weighted technique to AR models with AR errors. Further, Ref. [23] applied the empirical likelihood method to test the heteroscedasticity for errors of single-index model. Ref. [6] developed a unified empirical likelihood inference method to test the predictability regardless of the properties of the predicting variable. Ref. [24] considered the unified test problem in a predictive regression

model. To move the effect of the possible existence of an intercept, the idea of data-splitting has also been developed in [24]. The literature above inspired the current research.

We organize the rest of this paper as follows. Section 2 develops the unified test for the AR structure of the AR models. Section 3 reports the finite-sample simulation results. Section 4 applies the proposed test to the exchange rates between the U.S. dollar and eight countries. Section 5 concludes this paper. The detailed proof of the main theorem is specified in Appendix A.

## 2. Methodologies and Asymptotic Results

Supposing the random observations $\{X_t\}_{t=1}^n$ are generated from the model (2) with possible AR errors. Formulate $\boldsymbol{\psi} = (\boldsymbol{\psi}_1^\top, \boldsymbol{\psi}_2^\top)^\top$ and let $\boldsymbol{\psi}_0 = (\boldsymbol{\psi}_{1,0}^\top, \boldsymbol{\psi}_{2,0}^\top)^\top$ be its true value.

Note that when $\boldsymbol{\theta} = \boldsymbol{\theta}_0$, $\{e_t(\boldsymbol{\theta})\}$ is a sequence of iid variables, it is more efficient to construct a statistical procedure on $\sum_{t=p+1}^n e_t^2(\boldsymbol{\theta})$ than on $\sum_{t=p+1}^n (X_t - \mu - \phi X_{t-1})^2$, as discussed in [13], where

$$
\begin{aligned}
e_t(\boldsymbol{\theta}) &= \left( X_t - \mu - \phi X_{t-1} + \sum_{i=1}^p \psi_i (X_{t-i} - \mu - \phi X_{t-i-1}) \right), \quad \text{and} \\
\varepsilon_t(\boldsymbol{\theta}) &= X_t - \mu - \phi X_{t-1},
\end{aligned}
$$

for a given $\boldsymbol{\theta} = (\mu, \phi, \boldsymbol{\psi}^\top)^\top$. However, their method depends on an assumption that the structure of the AR errors has been correctly specified, which needs to be pretested in practice. This motivates us to consider the following hypothesis:

$$
\mathcal{H}_0 : \boldsymbol{\psi}_2 = \boldsymbol{\psi}_{2,0} \quad \textit{versus} \quad \mathcal{H}_1 : \boldsymbol{\psi}_2 \neq \boldsymbol{\psi}_{2,0}.
$$

Remarkably, when $\boldsymbol{\psi}_{2,0} = 0$, $\{\varepsilon_t\}$ is a sequence of iid errors.

Note that when $\boldsymbol{\theta}$ takes the true underlying value $\boldsymbol{\theta}_0$, we have

$$
E(\boldsymbol{Z}_t^*(\boldsymbol{\theta}) | \mathcal{F}_{t-1}) = 0, \quad \text{for } t = p+1, \cdots, n,
$$

where $\boldsymbol{Z}_t^*(\boldsymbol{\theta}) = (Z_{t,1}^*(\boldsymbol{\theta}), \dots, Z_{t,2+p}^*(\boldsymbol{\theta}))^\top$, $\mathcal{F}_t$ denotes the sigma field generated by $\{e_s : t \leq t\}$, and

$$
\begin{cases}
Z_{t,1}^*(\boldsymbol{\theta}) = e_t(\boldsymbol{\theta}) \\
Z_{t,2}^*(\boldsymbol{\theta}) = e_t(\boldsymbol{\theta})(X_{t-1} + \sum_{i=1}^p \psi_i X_{t+m-i-1}) \\
Z_{t,2+i}^*(\boldsymbol{\theta}) = e_t(\boldsymbol{\theta})(X_{t-i} - \mu - \phi X_{t-i-1}), \quad i = 1, \dots, p,
\end{cases}
$$

which can be obtained by taking the partial differential to the sum of least squares, i.e.,

$$
\sum_{t=p+1}^n \left( X_t - \mu - \phi X_{t-1} + \sum_{j=1}^p \psi_j (X_{t-j} - \mu - \phi X_{t-j-1}) \right)^2,
$$

with respect to $\boldsymbol{\psi}$. Then, similar to [21], one can use the profile empirical likelihood method to construct a test for hypothesis $\mathcal{H}_0$ based on $\{\boldsymbol{Z}_t^*(\boldsymbol{\theta})\}$.

However, following [22], it is easy to verify that the resulting test does not converge in distribution to a standard chi-squared variable because the quantity $\frac{1}{\sqrt{n}} \sum_{t=p+1}^n \boldsymbol{Z}_t^*(\boldsymbol{\theta}_0) \boldsymbol{Z}_t^*(\boldsymbol{\theta}_0)^\top$ does not converge in probability for Case (ii), i.e., $\mu = 0$ and $\phi = 1 + \frac{c}{n}$ for some nonzero constant $c$ (nearly integrated if $c \neq 0$, and unit root if $c = 0$). As an improvement, one may use the weighted technique developed in [22] to construct a weighted empirical likelihood-based test. Unfortunately, the resulting testing statistic still faces a similar problem in the optimization step during the process of profiling the redundant parameters; see a similar discussion in [25].

To overcome this problem, we propose the construction of the following empirical likelihood function for $\boldsymbol{\theta}$:

$$L(\boldsymbol{\theta}) = \sup\left\{ \prod_{t=p+1}^{m} m\delta_t : \delta_1 \geq 0, \cdots, \delta_m \geq 0, \sum_{t=p+1}^{m} \delta_t = 1, \sum_{t=p+1}^{m} \delta_t Z_t(\boldsymbol{\theta}) = 0 \right\},$$

based on the data-splitting idea, where $\boldsymbol{Z}_t(\boldsymbol{\theta}) = (Z_{t,1}(\boldsymbol{\theta}), \ldots, Z_{t,2+p}(\boldsymbol{\theta}))^\top$ with

$$\begin{cases} Z_{t,1}(\boldsymbol{\theta}) = e_t(\boldsymbol{\theta}) \\ Z_{t,2}(\boldsymbol{\theta}) = e_{t+m}(\boldsymbol{\theta}) \left( \frac{X_{t+m-1}}{\sqrt{1+X_{t+m-p-1}^2}} + \sum_{i=1}^{p} \psi_i \frac{X_{t+m-i-1}}{\sqrt{1+X_{t+m-p-i-1}^2}} \right) \\ Z_{t,2+i}(\boldsymbol{\theta}) = e_t(\boldsymbol{\theta})(X_{t-i} - \mu - \phi X_{t-i-1}), \quad i = 1, \ldots, p, \end{cases} \quad (3)$$

where $m = [n/2]$ with $[\cdot]$ is the floor function. That is, we use the second half of the data to handle $\phi$, and the first half of the data to handle the rest of the parameters. Here, $\sqrt{1 + X_{t+m-p-i-1}}$ is mainly used for technical consideration, which can relieve the correlation among $\{\boldsymbol{Z}_t(\boldsymbol{\theta})\}$, and consequently improve the finite sample performance of the EL test.

Since our aim is to test $\mathcal{H}_0$ related to $\boldsymbol{\psi}$, we are only interested in the parameter $\boldsymbol{\psi}$. To this end, we treat the other parameters as redundant parameters, as in [21], and obtain the profile empirical likelihood ratio as $\ell^p(\boldsymbol{\psi}) := \min_{\mu,\phi} \ell(\mu, \phi, \boldsymbol{\psi})$.

To derive the asymptotic result for $\ell^p(\boldsymbol{\psi})$, we need the following regular conditions:

- (C1) Suppose $\{X_t\}$ follows one of the following cases:
  - (i) (Stationary) $|\phi| < 1$, independent of $n$;
  - (ii) (Non-stationary without an intercept) $\phi = 1 - \frac{c}{n}$ for some constant $c$ independent of $n$ with $\mu = 0$;
  - (iii) (Non-stationary with an intercept) $\phi = 1 - \frac{c}{n}$ for some constant $c$ independent of $n$ with $\mu \neq 0$;
- (C2) $\psi(z) = 1 - \sum_{j=1}^{p} \psi_j z^j \neq 0$ when $|z| < 1$, and $\psi(z)$ has no common root with $\psi_p \neq 0$.
- (C3) $\{e_t\}$ are iid random errors, and satisfy $E(|e_t|^{2+\delta}) < \infty$ for some constant $\delta > 0$.

These conditions are quite common, and can be found in studies such as [13]. Here, (C2) is assumed to guarantee the stationarity of $\{\varepsilon_t\}$.

Under these conditions, we have the following result.

**Theorem 1.** *Suppose Conditions (C1)–(C3) hold. Then, under the null hypothesis $\mathcal{H}_0$,*

$$\ell^p(\boldsymbol{\psi}_0) \xrightarrow{d} \chi_p^2,$$

*as $n \to \infty$, where $\chi_p^2$ denotes a chi-squared random variable with p degrees of freedom, and '$\xrightarrow{d}$' denotes the convergence in distribution.*

**Remark 1.** *Using a similar proof to that of Theorem 1, we can show that*

$$\tilde{\ell}^p(\boldsymbol{\psi}_{0,2}) \xrightarrow{d} \chi_r^2, \text{ as } n \to \infty,$$

*where $\tilde{\ell}^p(\boldsymbol{\psi}_{0,2}) = \min_{\mu,\phi,\boldsymbol{\psi}_1} \ell(\mu, \phi, (\boldsymbol{\psi}_1^\top, \boldsymbol{\psi}_{0,2}^\top)^\top)$ with r being the dimension of $\boldsymbol{\psi}_{0,2}$, which is the true value.*

Theorem 1 is desirable because it shows that the proposed test has a standard chi-squared distribution asymptotically, regardless of which one of the Cases (i)–(iii) is followed by $\{X_t\}$.

Based on Theorem 1, we may reject the null hypothesis $\mathcal{H}_0$ once $\ell^p(\boldsymbol{\psi}_0) > \chi_r^2(1-a)$ at the significance level $a \in (0,1)$, where $\chi_r^2(1-a)$ denotes the $(1-a)$-th quantile of $\chi_r^2$.

## 3. Simulation Results

In this section, we conduct some simulations to investigate the finite sample performance of the proposed test in terms of both size and power. The simulations consist of three parts. In the first part, we investigate the finite sample performance of the proposed profile empirical likelihood, and compare it with a combination of the LB test and the Akaike information criterion (AIC), i.e., using firstly the LB test to detect whether there exists a serial correlation in the residuals, and then by employing the AIC to determine the order of the AR structure in residuals. In the second part, we investigate the possibility of using the proposed method to test whether or not $\boldsymbol{\psi}$ is equal to some given $\boldsymbol{\psi}_0$, which may be useful when verifying the extent of the stationarity of the AR errors. Note that the combination of the LB and AIC cannot be used to fulfill this type of task. In the last part, we study the impact of misdetermining the AR structure of the errors on the finite sample performance of the unit root test developed in [13]. The LB test is computed with the R function *Box.test.R*, while for the computing of the profile empirical likelihood, we first use R package *emplik* to obtain the log-empirical likelihood ratio, and then optimize this log ratio by using the *nlm.R* function. All of these R functions are well-documented, and are currently available from the CRAN of the R-project.

In the first part, the random observations $\{X_t\}$ are generated from the model (2) with $\mu \in \{0, 0.01\}$, which indicates that the model has no intercept and an intercept item, respectively. We take $\phi$ from $\{0.5, 1, 1-\frac{1}{n}\}$, where 0.5 indicates that $X_t$ is a stationary process, and 1 indicates that it is a unit root process, while $1-\frac{1}{n}$ indicates a near unit root process. $\{e_t\}$ is a sequence of iid random variables with means of zero and variances of one. $\{\varepsilon_t\}$ follows the three different scenarios listed below.

- **S1:** The null hypothesis $\mathcal{H}_0^{(1)} : \boldsymbol{\psi} = (0,0,0)^\top$, i.e., $\varepsilon_t$ has no serial correlation. The local alternative hypothesis: $\boldsymbol{\psi} = \boldsymbol{\psi}_0 = (d/\sqrt{n}, 0, 0)^\top$ for some $d > 0$.

- **S2:** The null hypothesis $\mathcal{H}_0^{(2)} : \boldsymbol{\psi} = (0.1, 0, 0)^\top$, i.e., $\varepsilon_t$ has first-order serial correlation. The local alternative hypothesis: $\boldsymbol{\psi} = \boldsymbol{\psi}_0 = (0.1, d/\sqrt{n}, 0)^\top$ for some $d > 0$.

- **S3:** The null hypothesis $\mathcal{H}_0^{(3)} : \boldsymbol{\psi} = (0.1, 0.1, 0)^\top$, i.e., $\varepsilon_t$ has second-order serial correlation. The local alternative hypothesis: $\boldsymbol{\psi} = \boldsymbol{\psi}_0 = (0.1, 0.1, d/\sqrt{n})^\top$ for some $d > 0$.

In all Scenarios **S1**–**S3**, $d$ is taken from $\{1, 3, 5, 7\}$. All computations are carried out 10,000 times with $n$ ranging from 300 to 1200.

Table 1 reports the size performance of the proposed method with different settings at the significance levels $\tau = 0.05$. We also report the ratios of determining the order of the AR error incorrectly by using the AIC of Scenarios **S1**–**S3** under the condition of $\mathcal{H}_0$ for comparison. The EL method has a good performance in all Scenarios **S1**–**S3**. The results show that the size values of the EL method gradually converge to the significance level as the sample size $n$ increases, no matter whether $X_t$ is a stationary process, a near unit root process, or a unit root process, and regardless of whether $\mu$ is 0 or not. Conversely, for the AIC method, when $X_t$ follows a stationary process, the ratios of determining the order of the AR error incorrectly are only closer to 5% in **S1**. Note that it performs poorly for the rest of the settings, meaning that it is affected greatly by the stationarity of $\{X_t\}$.

Figure 1 shows the power performance of the EL method. We can see that in **S1** and **S2**, when $X_t$ follows a stationary process, the convergence rate is the slowest. When $X_t$ follows a near unit root process or a unit root process, as the value of $d$ increases, the power converges quickly to 1. In **S3**, when $X_t$ follows a near unit root process or a unit root process, as the value of $d$ increases to 7, the power values have a slightly descending tendency. This implies that although the stationarity of $\{X_t\}$ does not impact on the order of the local alternative hypothesis, it does affect the power function of the EL method.

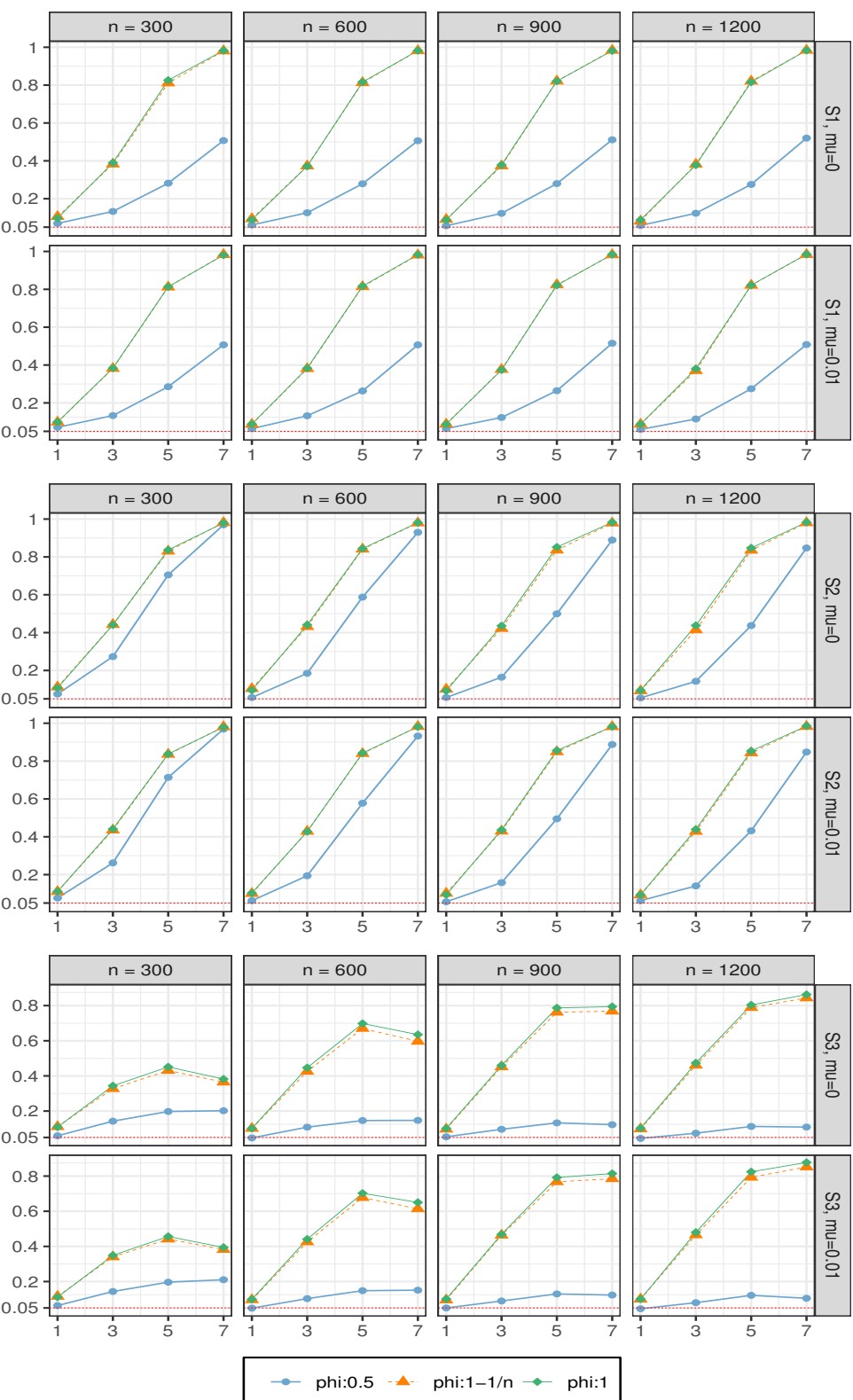

**Figure 1.** The power performance of EL method at $\tau = 0.05$.

**Table 1.** Empirical reject probabilities at $\tau = 0.05$.

| Scenarios | $\mu$ | $\phi$ | EL | | | | AIC | | | |
|---|---|---|---|---|---|---|---|---|---|---|
| | | | 300 | 600 | 900 | 1200 | 300 | 600 | 900 | 1200 |
| **S1** | 0 | 0.5 | 0.0601 | 0.0518 | 0.0541 | 0.0554 | 0.0715 | 0.0693 | 0.0762 | 0.0768 |
| | | $1-\frac{1}{n}$ | 0.0721 | 0.0659 | 0.0609 | 0.0600 | 0.2285 | 0.2308 | 0.2337 | 0.2404 |
| | | 1 | 0.0733 | 0.0657 | 0.0596 | 0.0597 | 0.2328 | 0.2337 | 0.2357 | 0.2396 |
| | 0.01 | 0.5 | 0.0699 | 0.0575 | 0.0509 | 0.0509 | 0.0760 | 0.0719 | 0.0732 | 0.0698 |
| | | $1-\frac{1}{n}$ | 0.0710 | 0.0611 | 0.0608 | 0.0588 | 0.2301 | 0.2323 | 0.2407 | 0.2301 |
| | | 1 | 0.0733 | 0.0578 | 0.0612 | 0.0581 | 0.2267 | 0.2330 | 0.2300 | 0.2406 |
| **S2** | 0 | 0.5 | 0.0587 | 0.0464 | 0.0469 | 0.0517 | 1.0000 | 1.0000 | 1.0000 | 1.0000 |
| | | $1-\frac{1}{n}$ | 0.0652 | 0.0620 | 0.0629 | 0.0570 | 0.5531 | 0.3458 | 0.2635 | 0.2243 |
| | | 1 | 0.0689 | 0.0587 | 0.0630 | 0.0563 | 0.5505 | 0.3450 | 0.2574 | 0.2246 |
| | 0.01 | 0.5 | 0.0591 | 0.0454 | 0.0481 | 0.0514 | 0.9999 | 1.0000 | 1.0000 | 1.0000 |
| | | $1-\frac{1}{n}$ | 0.0691 | 0.0546 | 0.0586 | 0.0607 | 0.5481 | 0.3394 | 0.2665 | 0.2339 |
| | | 1 | 0.0694 | 0.0589 | 0.0587 | 0.0578 | 0.5454 | 0.3463 | 0.2661 | 0.2254 |
| **S3** | 0 | 0.5 | 0.0557 | 0.0498 | 0.0474 | 0.0463 | 0.8969 | 0.8466 | 0.8131 | 0.7832 |
| | | $1-\frac{1}{n}$ | 0.0778 | 0.0613 | 0.0621 | 0.0590 | 0.5963 | 0.3284 | 0.2238 | 0.1732 |
| | | 1 | 0.0862 | 0.0635 | 0.0604 | 0.0601 | 0.5978 | 0.3318 | 0.2252 | 0.1789 |
| | 0.01 | 0.5 | 0.0550 | 0.0496 | 0.0483 | 0.0461 | 0.9005 | 0.8445 | 0.8195 | 0.7848 |
| | | $1-\frac{1}{n}$ | 0.0859 | 0.0692 | 0.0601 | 0.0587 | 0.6041 | 0.3332 | 0.2173 | 0.1832 |
| | | 1 | 0.0834 | 0.0646 | 0.0578 | 0.0585 | 0.5971 | 0.3318 | 0.2201 | 0.1812 |

In the second part, we consider testing whether or not $\boldsymbol{\psi}$ is equal to some given $\boldsymbol{\psi}_0$. We simulate two settings, i.e.,

- **(I):** The null hypothesis $\tilde{\mathcal{H}}_0^{(1)} : \boldsymbol{\psi} = (\psi_1, \psi_2) = (0.1, 0.3)^\top$ against the local alternative hypothesis: $\boldsymbol{\psi} = \boldsymbol{\psi}_0 = (0.1 + \frac{d}{\sqrt{n}}, 0.3 + \frac{d}{\sqrt{n}})^\top$, for some $d > 0$.

- **(II):** The null hypothesis $\tilde{\mathcal{H}}_0^{(2)} : \psi_2 = 0.3$, against the local alternative hypothesis: $\psi_2 = 0.3 + \frac{d}{\sqrt{n}}$, for some $d > 0$.

The other parameters are the same as those in the first part. The size ($d = 0$) and power ($d \in \{3, 5, 10\}$) performances are shown in Figure 2. As expected, similar observations can be found in Figure 3 as in the first part of simulations.

The simulation results in the first and second parts show that the proposed EL method has a good performance in specifying the AR error structure and testing whether or not $\boldsymbol{\psi}$ is equal to some given $\boldsymbol{\psi}_0$, thereby confirming the theoretical result obtained in Theorem 1. It is worth noting that when taking the AR error structure into account, accurate identification is crucial, because it will affect the unit root test of the AR model. Therefore, in the third part, we conduct the following simulation to show the benefit of conducting a predefined test.

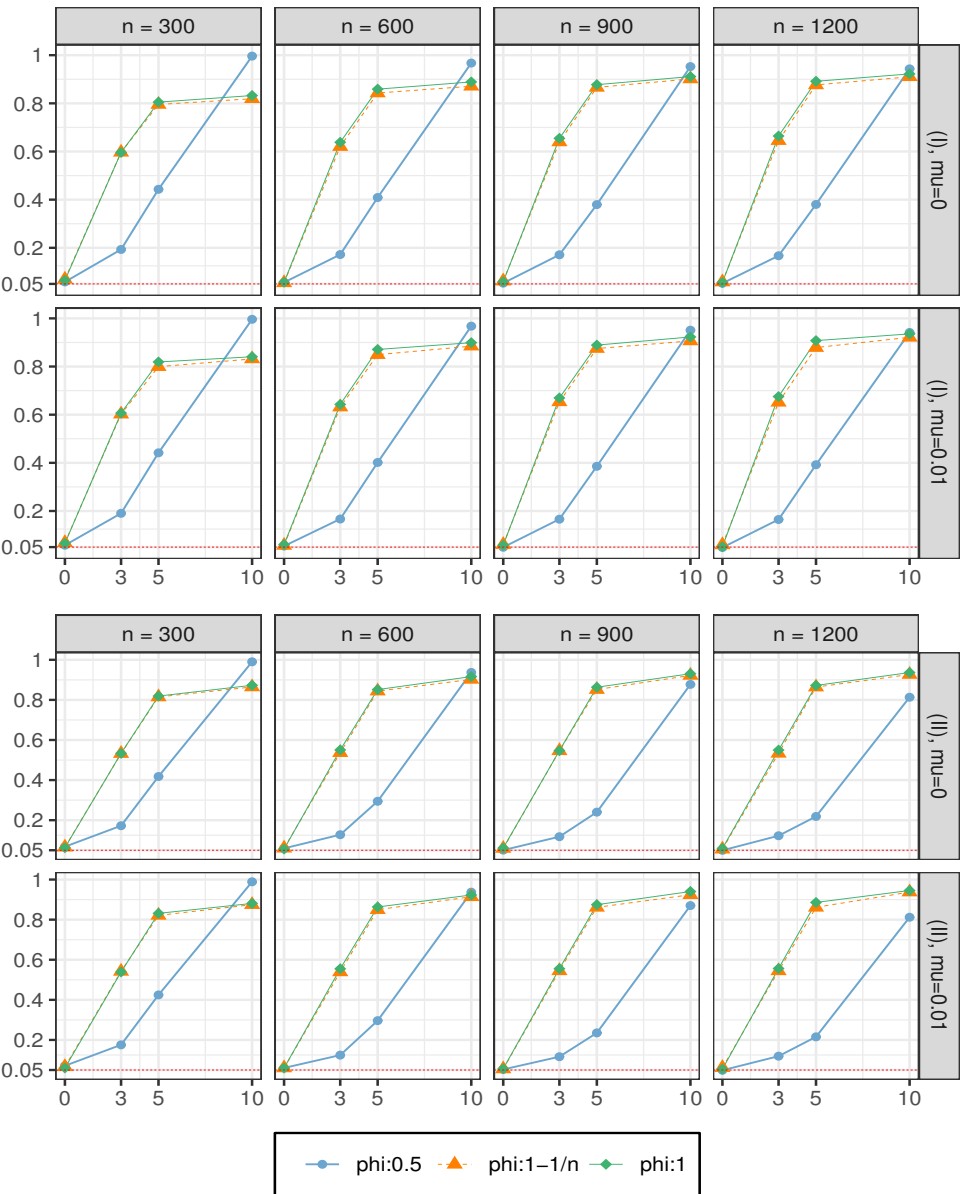

**Figure 2.** Empirical reject probabilities at $\tau = 0.05$.

Step 1: We generate an AR model with an AR(1) error structure, and the parameters are $\mu = 0, \phi = 1 - \frac{1}{n}, \psi = \frac{10}{\sqrt{n}}$. Then, we use the EL and AIC methods to determine the order of the AR error. We consider sample sizes of 600 and 1200, repeat the tests 10,000 times, and record the order determination counts under the two methods. The results are shown in Figure 3. The abscissa represents the order of the AR error, and the ordinate is the number of each order. It can be seen from Figure 3 that under all sample sizes, the two methods show that the residuals have a serial correlation. For the EL test, in 10,000 experiments, 9398 of them are correctly ordered, and the error rate is only 6.02%. When the sample size increases to 1200, the error rate decreases to 5.84%. For the AIC, when the sample size is 600, the error rate is 34.21%, and when the sample size is 1200, the error rate is 22.75%. It is obvious that compared with the AIC method, the EL test has advantages in identifying the order of the correlated errors, which is consistent with the above simulation results.

Step 2: We use the method proposed in [13] to test the unit root of an AR (1) model when the AR error order is correctly and incorrectly determined. Table 2 records the probability of identifying a unit root when the real data are a near unit root. The results show that when the true underlying structure of the AR error is incorrectly specified, the

power of the test proposed in [13] suffers from a loss compared to the case when the true underlying structure of the AR error is correctly specified. This shows the necessity of correctly testing the AR error structure before conducting the unit root test if one wants to obtain a more reliable unit root test result.

To summarize, the EL method proposed in this paper has obvious advantages in identifying the AR error structure, and these two methods are crucial in the subsequent real data analysis.

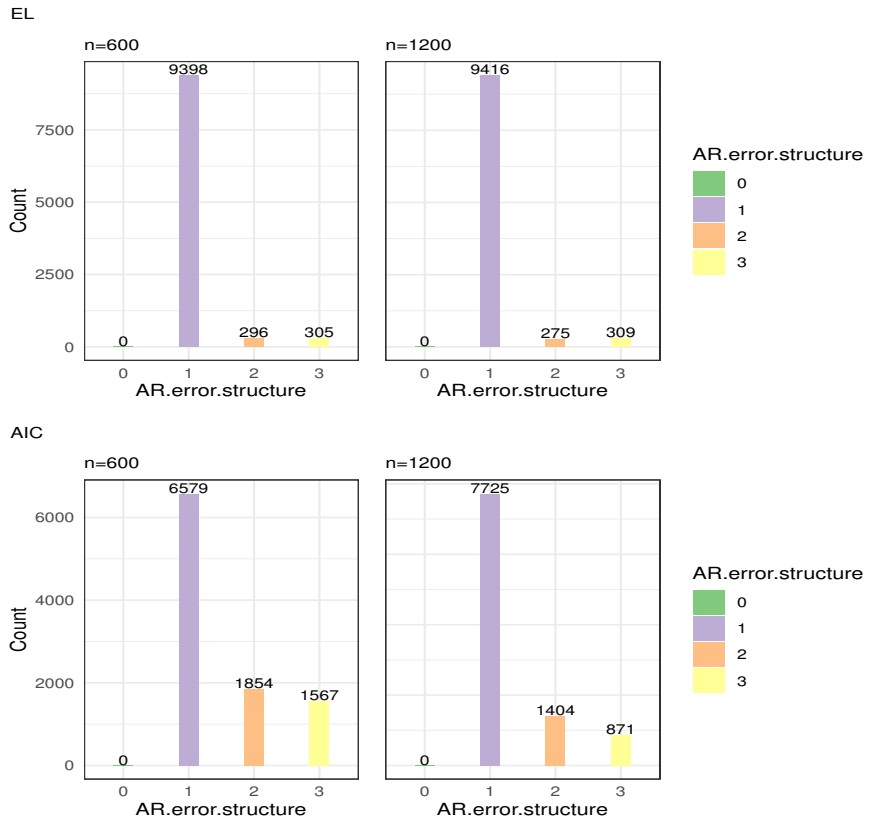

**Figure 3.** The results of the test for AR error structure between EL and AIC.

**Table 2.** The power performance of unit root test at $\tau = 0.05$.

| | **Right Order** | | | | **Wrong Order** | | | |
|---|---|---|---|---|---|---|---|---|
| $\phi$ | 300 | 600 | 900 | 1200 | 300 | 600 | 900 | 1200 |
| $1-\frac{1}{n}$ | 0.0474 | 0.0455 | 0.0422 | 0.0437 | 0.0438 | 0.0453 | 0.0419 | 0.0425 |
| $1-\frac{3}{n}$ | 0.1029 | 0.0941 | 0.0923 | 0.0862 | 0.0998 | 0.0918 | 0.0897 | 0.0851 |
| $1-\frac{5}{n}$ | 0.1795 | 0.1730 | 0.1718 | 0.1678 | 0.1747 | 0.1716 | 0.1666 | 0.1664 |
| $1-\frac{10}{n}$ | 0.4748 | 0.4596 | 0.4455 | 0.4367 | 0.4491 | 0.4496 | 0.4354 | 0.4307 |
| $1-\frac{15}{n}$ | 0.7620 | 0.7343 | 0.7159 | 0.6983 | 0.7276 | 0.7168 | 0.7028 | 0.6864 |

## 4. A Financial Real Data Application

In this section, we provide a real financial data example. The purpose of this section is to explore the error structure of different exchange rate markets. We collected the exchange rates of eight countries, including developed and developing countries, against the U.S. dollar. Currencies from developed countries include the Canadian dollar (CAD), Norwegian Kroner (NKR), Singapore dollar (SGD), Swedish Kronor (SKR) and Japanese yen (JPY). Currencies from developing countries include Chinese yuan (CNY), Thai baht (THB) and Sri Lanka Rupees (SRE). All data are downloaded from FRED database (*fred.stlouisfed.org*).

The sample period is the daily data from 2 January 2017 to 31 December 2020 ($n = 1044$). Their time series graphs are provide in Figure 4.

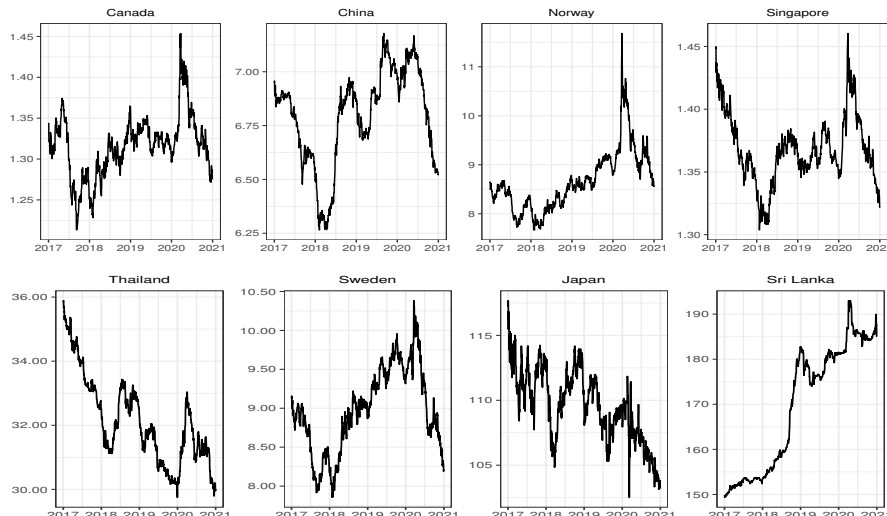

**Figure 4.** Time series graphs of 8 countries.

We report the least squares estimation of the unknown parameters $\mu$ and $\phi$, and the testing results of the EL, LB, and AIC methods, where the LB and AIC tests were conducted on residuals obtained from the least squares method. All results are listed in Table 3, in which the second and third columns are the estimated intercept and autoregressive coefficients, respectively; the fourth column is the order determination result of the EL test; the fifth column is the *p*-values of the EL test; and the last two columns are the *p*-values of LB test and the order determination result of AIC, respectively. The AIC test shows that most sequences have a serial correlation, except for CNY and JPY, while the EL method indicates that only one country's data has an AR error of up to an order of 2. Note that the AIC tends to determine the correlated errors with a higher order than for Cases (ii)–(iii), while for most cases, the estimated $\hat{\phi}$ is very close to 1, i.e., a near unit root. It seems that the testing results for this dataset coincide roughly with the observations in the simulations.

**Table 3.** Test results of 8 countries.

| Coutry | $\hat{\mu}$ | $\hat{\phi}$ | EL | *p*-Values | LB-Test | AIC |
|---|---|---|---|---|---|---|
| Canada | 0.0340 | 0.9992 | $AR(0)$ | 0.2122 | $7.7250 \times 10^{-3}$ *** | $AR(3)$ |
| China | 0.0169 | 0.9975 | $AR(0)$ | 0.8572 | 0.1391 | $AR(0)$ |
| Norway | 0.0462 | 0.9947 | $AR(0)$ | 0.4315 | $1.9450 \times 10^{-6}$ *** | $AR(2)$ |
| Singapore | 0.0146 | 0.9893 | $AR(0)$ | 0.8393 | 0.0238 ** | $AR(3)$ |
| Thailand | 0.1453 | 0.9953 | $AR(0)$ | 0.1666 | $4.0220 \times 10^{-3}$ *** | $AR(1)$ |
| Sweden | 0.0274 | 0.9968 | $AR(0)$ | 0.2165 | $5.6510 \times 10^{-3}$ *** | $AR(3)$ |
| Japan | 1.9372 | 0.9822 | $AR(0)$ | 0.2392 | 0.8896 | $AR(0)$ |
| Sri Lanka | 0.1738 | 0.9992 | $AR(2)$ | 0.2622 | $7.2230 \times 10^{-6}$ *** | $AR(2)$ |

Significance levels: * $p \leq 0.1$, ** $p \leq 0.05$, *** $p \leq 0.01$.

## 5. Conclusions

The AR model is widely used in time series data modeling. However, the direct application of AR models with iid errors is inadequate sometimes. A common practice is to further assume that AR models have errors of AR structure. Note that the relevant structure of the error affects the statistical inference of the AR model. Therefore, it is important to test the error structure of the model in both theoretical and practical analyses, which is not considered in the literature to the best of our knowledge. Motivated by this, this paper proposed a consistency empirical likelihood test method based on the idea of data splitting. The limit distribution of the EL statistic was proved to be chi-squared

asymptotically regardless of the process $\{X_t\}$ being stationary or non-stationary, and with or without an intercept term. The proof is challenging and different from that of the traditional profile empirical likelihood in [21], as the quantity $\frac{1}{\sqrt{n}} \sum_{t=p+1}^{n} \mathbf{Z}_t(\boldsymbol{\theta}_0)$ does not converge in distribution to a normally distributed vector. Fortunately, the limit distribution of the profile empirical likelihood-based test is still chi-squared because of the special block structure of the asymptotic covariance matrix. The simulation results illustrated that the proposed method could not only have a good performance in specifying the AR error structure, but also could sufficiently test whether the coefficients of the error item are equal to some given values, which can not be achieved by some existing serial correlation tests in the literature. The technique in the proof of Theorem 1 is challenging as in Case (ii) the theoretical proof involves handling the convergence in space, and the special block structure of the asymptotic covariance matrix. Hopefully, it is of potential usage in practice as it is difficult to detect whether the process $\{X_t\}$ is a unit root or near unit root process. Note that it is not necessary to make clear which case of (i)–(iii) the $\{X_t\}$ follows when using our proposed test in practice.

As noted by an an anonymous reviewer, an issue of interest is whether the current result can be extended to the case when $\varepsilon_t$ follows a autoregressive moving average model. Note that the discussion in this paper involves nonstationary Cases (ii) and (iii); it seems challenging to derive the related theoretical results. We will further consider this in the future.

**Author Contributions:** Conceptualization, X.L.; Methodology, X.L., X.W. and Y.F.; Software, X.W.; Validtion, X.L., Q.L. and L.T.; Formal analysis, X.W. and Y.F.; Investigation, Q.L. and L.T.; Resources, X.L.; Data curation, X.W.; Writing—original draft preparation, X.W., X.L., Y.F., L.T. and Q.L.; Writing—review and editing, X.L. and X.W.; Visualization, Q.L. and L.T.; Supervision, Q.L. and L.T.; Project administration, X.L.; Funding acquisition, X.L. All authors have read and agreed to the published version of the published version of the manuscript.

**Funding:** Xiaohui Liu's research was funded by NSF of China grant number 11971208, National Social Science Foundation of China grant number 21&ZD152 and the Outstanding Youth Fund Project of the Science and Technology Department of Jiangxi Province grant number 20224ACB211003. Li Tan's research was funded by Natural Science Foundation of China grant number 12061034 and China Postdoctoral Science Foundation grant number 2022M711424. Qing Liu's research was funded by NSF of Jiangxi Province grant number 20192BAB201005, China Post doctoral Science Foundation grant number 2020M671961, and Post doctoral Science Foundation of Jiangxi Province grant number 2019KY47 and The APC was funded by Xiaohui Liu.

**Data Availability Statement:** The real data used in the artical can be found at *fred.stlouisfed.org*.

**Acknowledgments:** We thank the three anonymous reviewers, the Associate Editor, and the Guest Editor of this Special Issue for their insightful comments, which have led to many improvements in this paper. Xiaohui Liu's research is supported by NSF of China (Grant No. 11971208), National Social Science Foundation of China (21&ZD152) and the Outstanding Youth Fund Project of the Science and Technology Department of Jiangxi Province (No. 20224ACB211003). Li Tan's research is supported by Natural Science Foundation of China (No. 12061034) and China Postdoctoral Science Foundation (No. 2022M711424). Qing Liu's research is supported by NSF of Jiangxi Province (No. 20192BAB201005), China Post doctoral Science Foundation (No. 2020M671961), and Post doctoral Science Foundation of Jiangxi Province (No. 2019KY47).

**Conflicts of Interest:** The authors declare no conflict of interest.

## Appendix A. Proof of the Main Result

In this appendix, we provide the detailed proofs for the main results. Before proceeding further, we need to first provide some necessary lemmas. For convenience, denote $\mu_0$, $\phi_0$, and $\boldsymbol{\psi}_0 := (\psi_{1,0}, \psi_{2,0}, \cdots, \psi_{p,0})^{\top}$ as the true values of $\mu$, $\phi$, and $\boldsymbol{\psi} := (\psi_1, \psi_2, \cdots, \psi_p)^{\top}$, respectively. Write $\boldsymbol{\theta}_0 := (\mu_0, \phi_0, \boldsymbol{\psi}_0^{\top})^{\top}$, and let $\mathcal{F}_t$ be the sigma field generated by $\{e_s : 1 \leq s \leq t, m+1 \leq s \leq m+t\}$.

For convenience, write $\boldsymbol{S}_t := (S_{t,1}, \tilde{\boldsymbol{S}}_t^\top)^\top = (S_{t,1}, S_{t,2}, \cdots, S_{t,p+1})^\top$, where

$$S_{t,1} = \frac{1}{\sqrt{m}} \sum_{i=1}^t e_i, \quad S_{t,2} = \frac{1}{\sqrt{m}} \sum_{i=1}^t e_i \varepsilon_{i-1}, \quad \cdots, \quad S_{t,p+1} = \frac{1}{\sqrt{m}} \sum_{i=1}^t e_i \varepsilon_{i-p},$$

for $t = 1, 2, \cdots, n$. By following [2], it is easy to check for any $s \in (0, 2]$ that

$$\boldsymbol{S}_{[ns]} \Rightarrow \boldsymbol{W}(s) := (W_e(s), \tilde{\boldsymbol{W}}(s)^\top)^\top = (W_e(s), \tilde{W}_1(s), \cdots, \tilde{W}_p(s))^\top, \tag{A1}$$

under Case (ii) as $n \to \infty$, where '$\Rightarrow$' denotes the convergence in space $D(0, 2]$ which is the space of real-valued functions of the interval $(0, 2]$ that are right continuous and have finite left limits, $[\cdot]$ denotes the floor function, and $\boldsymbol{W}(s)$ is a vector of Gaussian processes with covariance matrix $diag\{\sigma_e^2, \Sigma_{22}\}$ with $\sigma_e^2 = E(e_1^2)$ and

$$\Sigma_{22} = \begin{pmatrix} \sigma_e^2 E(\varepsilon_1^2) & \cdots & \sigma_e^2 E(\varepsilon_1 \varepsilon_p) \\ \vdots & \ddots & \vdots \\ \sigma_e^2 E(\varepsilon_p \varepsilon_1) & \cdots & \sigma_e^2 E(\varepsilon_p^2) \end{pmatrix}.$$

**Lemma A1.** *Under the same conditions of Theorem 1, as $n \to \infty$, we obtain*

- *For Case (i),*

$$\frac{1}{\sqrt{m}} \sum_{t=p+1}^m \boldsymbol{Z}_t(\boldsymbol{\theta}_0) \xrightarrow{d} N(0, \Sigma), \tag{A2}$$

*where '$\xrightarrow{p}$' denotes the convergence in probability, and $\Sigma = diag\{\Sigma_{11}, \Sigma_{22}\}$ and*

$$\Sigma_{11} = \begin{pmatrix} \sigma_e^2 & 0 \\ 0 & \sigma_e^2 \cdot \lim_{t \to \infty} E\left( \frac{X_{t+m-1}}{\sqrt{1+X_{t+m-p-1}^2}} + \sum_{i=1}^p \psi_i \frac{X_{t+m-i-1}}{\sqrt{1+X_{t+m-p-i-1}^2}} \right)^2 \end{pmatrix}.$$

- *For Case (ii),*

$$\frac{1}{\sqrt{m}} \sum_{t=p+1}^m \boldsymbol{Z}_t(\boldsymbol{\theta}_0) \tag{A3}$$
$$= (W_e(1), \omega_{2m}^* W_e(2) - \omega_m^* W_e(1), \tilde{W}_1(1), \cdots, \tilde{W}_p(1))^\top + o_p(1),$$

*where, for $k = m, 2m$,*

$$\omega_k^* = \left( \frac{X_{k-p-1}}{\sqrt{1+X_{k-p-1}^2}} + \sum_{i=1}^p \psi_i \frac{X_{k-p-i-1}}{\sqrt{1+X_{k-p-i-1}^2}} \right).$$

- *For Case (iii),*

$$\frac{1}{\sqrt{m}} \sum_{t=p+1}^m \boldsymbol{Z}_t(\boldsymbol{\theta}_0) \xrightarrow{d} N(0, \tilde{\Sigma}), \tag{A4}$$

*where $\tilde{\Sigma} = diag\{\sigma_e^2, \sigma_e^2, \Sigma_{22}\}$.*
*for Case (i), or $\Sigma_{11} = diag\{\sigma_e^2, \sigma_e^2\}$ for Cases (ii) and (iii). Note that $\sigma_e^2 = E(e_t^2)$.*

**Proof of Lemma A1.** Put $e_t = e_t(\boldsymbol{\theta}_0)$ and $\varepsilon_t = (X_{t-i} - \mu_0 - \phi_0 X_{t-i-1})$ for $i = 1, 2, \cdots, p$ when $\boldsymbol{\theta} = \boldsymbol{\theta}_0$. Then, it is easy to check that $\{\boldsymbol{Z}_t(\boldsymbol{\theta}_0)\}$ is a martingale difference sequence (MDS) with respect to the filter $\{\mathcal{F}_t\}$.

Next, for Case (i), under the conditions of Theorem 1, both $\{X_t\}$ and $\{\varepsilon_t\}$ are strictly stationary. Hence, for any $\boldsymbol{a} := (a_1, a_2, \cdots, a_{p+2})^\top \in R^{p+2}$, we obtain

$$\frac{1}{m} \sum_{t=p+1}^{m} E\left( (\boldsymbol{a}^\top \boldsymbol{Z}_t(\boldsymbol{\theta}_0))^2 | \mathcal{F}_{t-1} \right) \tag{A5}$$

$$= \boldsymbol{a}^\top \left( \frac{1}{m} \sum_{t=p+1}^{m} E\left( \boldsymbol{Z}_t(\boldsymbol{\theta}_0) \boldsymbol{Z}_t^\top(\boldsymbol{\theta}_0) | \mathcal{F}_{t-1} \right) \right) \boldsymbol{a}$$

$$\xrightarrow{p} \boldsymbol{a}^\top \Sigma \boldsymbol{a},$$

as $n \to \infty$ by using the law of large number for MDS [26]. Similarly, for any arbitrarily small $\epsilon > 0$, we obtain

$$\frac{1}{m} \sum_{t=p+1}^{m} E\left( |\boldsymbol{a}^\top \boldsymbol{Z}_t(\boldsymbol{\theta}_0)|^2 I(|\boldsymbol{a}^\top \boldsymbol{Z}_t(\boldsymbol{\theta}_0)| \geq \epsilon\sqrt{m}) | \mathcal{F}_{t-1} \right) \tag{A6}$$

$$\leq \frac{1}{\epsilon^{\delta_1} n^{\delta_1/2}} \frac{1}{m} \sum_{t=p+1}^{m} E\left( |\boldsymbol{a}^\top \boldsymbol{Z}_t(\boldsymbol{\theta}_0)|^{2+\delta_1} | \mathcal{F}_{t-1} \right)$$

$$\leq \frac{(p+2)^{1+\delta_1} \|\boldsymbol{a}\|^{2+\delta_1}}{\epsilon^{\delta_1} n^{\delta_1/2}} \frac{1}{m} \sum_{t=p+1}^{m} E\left( \sum_{i=1}^{p+2} |Z_{t,i}(\boldsymbol{\theta}_0)|^{2+\delta_1} | \mathcal{F}_{t-1} \right)$$

$$\xrightarrow{p} 0,$$

by noting that $E(|e_t|^{2+\delta_1}) < \infty$, which implies $E(|\varepsilon_t|^{2+\delta_1}) < \infty$ and in turn $E(|X_t|^{2+\delta_1}) < \infty$ when $\{\varepsilon_t\}$ and $\{X_t\}$ are strictly stationary. (A5) and (A6) together show the normality for Case (i) by using the central limit theorem for MDS [26].

For Case (ii), note that

$$X_{t+m-i-1} = \sum_{k=1}^{p} \phi^{k-1} \varepsilon_{t+m-i-k+1} + \phi^p X_{t+m-i-p-1}, \quad i = 1, 2, \ldots, p,$$

and by [2,27], it holds that for any $s \in (0, 1]$

$$\frac{X_{[ns]}}{\sqrt{n}} \Rightarrow J_c(s) := \int_0^s e^{-c(s-r)} dW(r) \text{ in the space } D((0, 1]), \tag{A7}$$

as $n \to \infty$. Using these, it is easy to check that, as $n \to \infty$,

$$\frac{1}{\sqrt{m}} \sum_{t=p+1}^{m} e_{t+m} \left( \frac{X_{t+m-1}}{\sqrt{1 + X_{t+m-p-1}^2}} + \sum_{i=1}^{p} \psi_i \frac{X_{t+m-i-1}}{\sqrt{1 + X_{t+m-p-i-1}}} \right)$$

$$= \frac{1}{\sqrt{m}} \sum_{t=p+1}^{m} e_{t+m} \left( \frac{X_{t+m-p-1}}{\sqrt{1 + X_{t+m-p-1}^2}} + \sum_{i=1}^{p} \psi_i \frac{X_{t+m-p-i-1}}{\sqrt{1 + X_{t+m-p-i-1}}} \right) + o_p(1).$$

Note that

$$\frac{1}{\sqrt{m}} \sum_{t=p+1}^{m} e_{t+m} \left( \frac{X_{t+m-p-1}}{\sqrt{1 + X_{t+m-p-1}^2}} + \sum_{i=1}^{p} \psi_i \frac{X_{t+m-p-i-1}}{\sqrt{1 + X_{t+m-p-i-1}}} \right)$$

$$:= \frac{1}{\sqrt{m}} \sum_{t=p+1}^{m} e_{t+m} \omega_{t+m}^*$$

$$= \sum_{t=p+1}^{m} (S_{t+m,1} - S_{t+m-1,1}) \omega_{t+m}^*$$

$$= S_{2m,1} \omega_{2m}^* - S_{p+m,1} \omega_{p+m+1}^* + \sum_{t=p+1}^{m-1} S_{t+m,1} (\omega_{t+m}^* - \omega_{t+m+1}^*).$$

Using (A7), we obtain $w_{2m} \xrightarrow{p} \text{sgn}(J_c(2))$, where $\text{sgn}(\cdot)$ denotes the sign function. As $X_{[ns]} = O_p(\sqrt{m})$ and $n \to \infty$ for Case (ii), it is easy to check that there exists some $d \in (0, \frac{1}{2})$, for $i = 0, 1, \cdots, p$, such that

$$\left| \sum_{t=p+1}^{m-1} S_{t+m,1} \left( \frac{X_{t+m-p-i-1}}{\sqrt{1 + X_{t+m-p-i-1}}} - \frac{X_{t+m-p-i}}{\sqrt{1 + X_{t+m-p-i}}} \right) \right|$$

$$= \left| \sum_{t=p+1}^{m-1} S_{t+m,1} \frac{X_{t+m-p-i} - X_{t+m-p-i-1}}{(1 + \xi_{t,i,*}^2)^{\frac{3}{2}}} \right|$$

$$\leq O_p(m^{-d}) \times \frac{1}{m} \sum_{t=p+1}^{m-1} \{|S_{t+m,1}||X_{t+m-p-i} - X_{t+m-p-i-1}|\} = o_p(1),$$

where $\xi_{t,i,*}$ lies between $X_{t+m-p-i}$ and $X_{t+m-p-i-1}$. This shows

$$\frac{1}{\sqrt{m}} \sum_{t=p+1}^{m} \mathbf{Z}_t(\boldsymbol{\theta}_0)$$

$$= (S_{m,1}, S_{2m,1} \omega_{2m}^* - S_{p+m,1} \omega_{p+m+1}^*, S_{m,2}, \cdots, S_{m,p+1})^\top + o_p(1), \quad \text{as } n \to \infty.$$

Then, the asymptotic result for Case (ii) follows immediately based on (A1).

Case (iii) can be proved similarly as Cases (i) and (ii). We omit the details.  □

**Lemma A2.** *Under the same conditions of Theorem 1, as $n \to \infty$, we find that*

- *(a) $\frac{1}{\sqrt{m}} \sum_{t=p+1}^{m} \mathbf{Z}_t(\boldsymbol{\theta}_*) = \frac{1}{\sqrt{m}} \sum_{t=p+1}^{m} \mathbf{Z}_t(\boldsymbol{\theta}_0) + O_p(1)$, uniformly for $(\mu, \phi) \in \mathcal{B}$,*
- *(b) $\frac{1}{m} \sum_{t=p+1}^{m} \mathbf{Z}_t(\boldsymbol{\theta}_*) \mathbf{Z}_t^\top(\boldsymbol{\theta}_*) = \frac{1}{m} \sum_{t=p+1}^{m} \mathbf{Z}_t(\boldsymbol{\theta}_0) \mathbf{Z}_t^\top(\boldsymbol{\theta}_0) + o_p(1) = \Sigma_0 + o_p(1)$, uniformly for $(\mu, \phi) \in \mathcal{B}$,*
- *(c) $\max_{p+1 \leq t \leq m} \sup_{\mathcal{B}} \|\mathbf{Z}_t(\boldsymbol{\theta}_*)\| = o_p(\sqrt{m})$,*

*where $\boldsymbol{\theta}_* = (\mu, \phi, \boldsymbol{\psi}_0^\top)^\top$,*

$$\mathcal{B} = \begin{cases} \{(\mu, \phi) : |\mu - \mu_0| + |\phi - \phi_0| < C/\sqrt{m}\} & \text{for Case (i)}, \\ \{(\mu, \phi) : |\mu - \mu_0| + \sqrt{m}|\phi - \phi_0| < C/\sqrt{m}\} & \text{for Case (ii)}, \\ \{(\mu, \phi) : |\mu - \mu_0| + m|\phi - \phi_0| < C/\sqrt{m}\} & \text{for Case (iii)}, \end{cases}$$

*for some positive constant $C$, $\Sigma_0 = \text{diag}\{\Sigma_{11}, \Sigma_{22}\}$ for Case (i), and $\text{diag}\{\sigma_e^2, \sigma_e^2, \Sigma_{22}\}$ for Cases (ii) and (iii).*

**Proof of Lemma A2.** We only prove Parts (a) and (c), as the proof of (b) is trivial based on those of (a) and (c).

For Part (a), note that

$$
\begin{aligned}
e_t(\boldsymbol{\theta}_*) - e_t &= X_t - \mu - \phi X_{t-1} + \sum_{i=1}^{p} \psi_{i,0}(X_{t-i} - \mu - \phi X_{t-i-1}) \\
&\quad - \{X_t - \mu_0 - \phi_0 X_{t-1} + \sum_{i=1}^{p} \psi_{i,0}(X_{t-i} - \mu_0 - \phi_0 X_{t-i-1})\} \\
&= -(\mu - \mu_0)(1 + \sum_{i=1}^{p} \psi_{i,0}) - (\phi - \phi_0)\{X_{t-1} + \sum_{i=1}^{p} \psi_{i,0} X_{t-i-1}\},
\end{aligned}
$$

and $(X_{t-i} - \mu - \phi X_{t-i-1}) - \varepsilon_{t-i} = (\mu - \mu_0) + (\phi - \phi_0)X_{t-i-1}$ for any $t = p+1, \cdots, 2m$ and $i = 0, 1, \cdots, p$. Hence, we have

$$
\begin{aligned}
\sup_{\mathcal{B}} &\left| \frac{1}{\sqrt{m}} \sum_{t=p+1}^{m} (Z_{t,1}(\boldsymbol{\theta}_*) - Z_{t,1}(\boldsymbol{\theta}_0)) \right| \\
\leq\ &\sup_{\mathcal{B}} \left| -\sqrt{m}(\mu - \mu_0)(1 + \sum_{i=1}^{p} \psi_{i,0}) \right| \\
&\sup_{\mathcal{B}} \left| -m(\phi - \phi_0) \frac{1}{m\sqrt{m}} \sum_{t=p+1}^{m} \{X_{t-1} + \sum_{i=1}^{p} \psi_{i,0} X_{t-i-1}\} \right| \\
=\ &C \cdot \left| (1 + \sum_{i=1}^{p} \psi_{i,0}) \right| \cdot \left\{ 1 + \left| \int_0^1 J_c(s)ds \right| \right\} \\
=\ &O_p(1), \text{ as } n \to \infty.
\end{aligned}
$$

The proofs of $\sup_{\mathcal{B}} \left| \frac{1}{\sqrt{m}} \sum_{t=p+1}^{m}(Z_{t,k}(\boldsymbol{\theta}_*) - Z_{t,k}(\boldsymbol{\theta}_0)) \right|$, for $k = 2, \cdots, p+2$, follow a similar fashion. This shows Part (a).

For Part (c), based on the decomposition of $e_t(\boldsymbol{\theta}_*)$ given in Part (a), we similarly have

$$
\begin{aligned}
\max_{p+1 \leq t \leq m} \sup_{\mathcal{B}} |Z_{t,1}(\boldsymbol{\theta}_*)| \leq\ &\max_{p+1 \leq t \leq m} |e_t| + |1 + \sum_{i=1}^{p} \psi_{i,0}| \cdot \sup_{\mathcal{B}} |\mu - \mu_0| \\
&+ \sup_{\mathcal{B}} |\phi - \phi_0| \cdot \max_{p+1 \leq t \leq m} |X_{t-1} + \sum_{i=1}^{p} \psi_{i,0} X_{t-i-1}| \\
=\ &o_p(\sqrt{m}),
\end{aligned}
$$

by using the Markov inequality based on the conditions of Theorem 1 as $n \to \infty$. $\max_{p+1 \leq t \leq m} \sup_{\mathcal{B}} |Z_{t,k}(\boldsymbol{\theta}_*)| = o_p(\sqrt{m})$, $k = 2, \cdots, p+2$, can be proved similarly. We omit the details. This shows Part (c). $\square$

**Proof of Theorem 1.** In the following, we only prove Case (ii), as Cases (i) and (iii) follow a similar fashion.

Based on Lemmas A1 and A2, we can show by using similar techniques as in Theorem 1 of [28] that

$$
\ell(\mu, \phi, \boldsymbol{\psi}_0) = \left( \frac{1}{\sqrt{m}} \sum_{t=p+1}^{m} \boldsymbol{Z}_t(\boldsymbol{\theta}_*) \right)^{\top} \Sigma_0^{-1} \left( \frac{1}{\sqrt{m}} \sum_{t=p+1}^{m} \boldsymbol{Z}_t(\boldsymbol{\theta}_*) \right) + o_p(1), \tag{A8}
$$

uniformly for $(\mu, \phi) \in \mathcal{B}$. Note that $(\mu_0, \phi_0) \in \mathcal{B}$. Trivially, with (A9), it follows

$$
\ell(\mu_0, \phi_0, \boldsymbol{\psi}_0) = \left( \frac{1}{\sqrt{m}} \sum_{t=p+1}^{m} \boldsymbol{Z}_t(\boldsymbol{\theta}_0) \right)^{\top} \Sigma_0^{-1} \left( \frac{1}{\sqrt{m}} \sum_{t=p+1}^{m} \boldsymbol{Z}_t(\boldsymbol{\theta}_0) \right) + o_p(1), \tag{A9}
$$

as $n \to \infty$.

$\varepsilon_t(\mu, \phi) = X_t - \mu - \phi X_{t-1}$, $\widetilde{X}_{t-1} = X_{t-1} + \sum_{i=1}^p \psi_{i,0} X_{t-i-1}$ and $\gamma_0 = 1 + \sum_{i=1}^p \psi_{i,0}$.
Note that, with (A7), we have

$$\frac{1}{m} \sum_{t=p+1}^m \{\frac{1}{\sqrt{m}} \widetilde{X}_{t-1}\} \xrightarrow{d} \gamma_0 \int_0^1 J_c(s) ds,$$

$$\frac{1}{m} \sum_{t=p+1}^m \{\frac{1}{\sqrt{m}} \widetilde{X}_{t+m-1} \omega_{t+m}^*\} \xrightarrow{d} \gamma_0 \int_1^2 \text{sgn}(J_c(s)) ds, \text{ as } n \to \infty.$$

Next, since $\varepsilon_t = \Phi^{-1}(B) e_t$, where $\Phi(B) = 1 + \sum_{i=1}^p \psi_{i,0} B^i$ with $B$ being the lag operator satisfying $B^j e_t = e_{t-j}$, which is a linear process of $\{e_t\}$, we may show that

$$\frac{1}{\sqrt{m}} \sum_{t=p+1}^m \{\frac{1}{\sqrt{m}} X_{t-1} e_t\} = O_p(1), \quad \text{and} \quad \frac{1}{\sqrt{m}} \sum_{t=p+1}^m \{\frac{1}{\sqrt{m}} X_{t-1} \varepsilon_t\} = O_p(1), \text{ as } n \to \infty,$$

based on the martingale decomposition as those maintained in [29,30]; see the proof of Theorem 3.1 of [31] for similar discussions. Then, we have

$$\frac{1}{\sqrt{m}} \sum_{t=p+1}^m \{Z_t(\theta_*) - Z_t(\theta_0)\}$$

$$= -\frac{1}{\sqrt{m}} \sum_{t=p+1}^m \begin{pmatrix} (\mu - \mu_0)\gamma_0 + (\phi - \phi_0)\widetilde{X}_{t-1} \\ \{(\mu - \mu_0)\gamma_0 + (\phi - \phi_0)\widetilde{X}_{t+m-1}\}\omega_{t+m}^* \\ \{(\mu - \mu_0)\gamma_0 + (\phi - \phi_0)\widetilde{X}_{t-1}\}\varepsilon_{t-1}(\mu, \phi) \\ \cdots \\ \{(\mu - \mu_0)\gamma_0 + (\phi - \phi_0)\widetilde{X}_{t-1}\}\varepsilon_{t-p}(\mu, \phi) \end{pmatrix}$$

$$- \frac{1}{\sqrt{m}} \sum_{t=p+1}^m \begin{pmatrix} 0 \\ 0 \\ e_t((\mu - \mu_0) + (\phi - \phi_0)X_{t-2}) \\ \cdots \\ e_t((\mu - \mu_0) + (\phi - \phi_0)X_{t-p-1}) \end{pmatrix}$$

$$= -\begin{pmatrix} \gamma_0 & \gamma_0 \int_0^1 J_c(s) ds \\ \gamma_0 & \gamma_0 \int_1^2 \text{sgn}(J_c(s)) ds \\ 0 & 0 \\ \ddots & \ddots \\ 0 & 0 \end{pmatrix} \begin{pmatrix} \sqrt{m}(\mu - \mu_0) \\ m(\phi - \phi_0) \end{pmatrix} + o_p(1)$$

$$:= \Gamma \begin{pmatrix} \sqrt{m}(\mu - \mu_0) \\ m(\phi - \phi_0) \end{pmatrix} + o_p(1), \tag{A10}$$

uniformly for $(\mu, \phi) \in \mathcal{B}$ as $n \to \infty$.

Based on (A10), it is then easy to check that the minimizer, say $(\hat{\mu}, \hat{\phi})$, of $\ell(\mu, \phi, \boldsymbol{\psi}_0) - \ell(\mu_0, \phi_0, \boldsymbol{\psi}_0)$ must be in $\mathcal{B}$, and satisfies

$$\begin{pmatrix} \sqrt{m}(\hat{\mu} - \mu_0) \\ m(\hat{\phi} - \phi_0) \end{pmatrix} = -(\Gamma^\top \Sigma_0^{-1} \Gamma)^{-1} \Gamma^\top \Sigma_0^{-1} \cdot \frac{1}{\sqrt{m}} \sum_{t=p+1}^m Z_t(\theta_0) + o_p(1), \text{ as } n \to \infty.$$

Then, it follows

$$\ell(\hat{\mu}, \hat{\phi}, \boldsymbol{\psi}_0)$$

$$= \left(\frac{1}{\sqrt{m}} \sum_{t=p+1}^m Z_t(\theta_0)\right)^\top \left(\Sigma_0^{-1} - \Sigma_0^{-1} \Gamma (\Gamma^\top \Sigma_0^{-1} \Gamma)^{-1} \Gamma^\top \Sigma_0^{-1}\right) \frac{1}{\sqrt{m}} \sum_{t=p+1}^m Z_t(\theta_0) + o_p(1).$$

Further note that

$$\Gamma(\Gamma^\top \Sigma_0^{-1} \Gamma)^{-1} \Gamma^\top = \Gamma(\Gamma_1^\top \Sigma_{11}^{-1} \Gamma_1)^{-1} \Gamma = \begin{pmatrix} \Sigma_{11}^{-1} & \\ & 0 \end{pmatrix},$$

where $\Gamma_1 = \begin{pmatrix} \gamma_0 & \gamma_0 \int_0^1 J_c(s) ds \\ \gamma_0 & \gamma_0 \int_1^2 \text{sgn}(J_c(s)) ds \end{pmatrix}$. Then, we have

$$\Sigma_0^{-1} - \Sigma_0^{-1} \Gamma(\Gamma^\top \Sigma_0^{-1} \Gamma)^{-1} \Gamma^\top \Sigma_0^{-1} = \Sigma_0^{-1} - \Sigma_0^{-1} \begin{pmatrix} \Sigma_{11}^{-1} & \\ & 0 \end{pmatrix} \Sigma_0^{-1} = \begin{pmatrix} 0 & \\ & \Sigma_{22}^{-1} \end{pmatrix}.$$

Hence,

$$
\begin{aligned}
\ell^p(\boldsymbol{\psi}_0) &= \ell(\hat{\mu}, \hat{\phi}, \boldsymbol{\psi}_0) \\
&= \left( \frac{1}{\sqrt{m}} \sum_{t=p+1}^m \mathbf{Z}_t(\boldsymbol{\theta}_0) \right)^\top \begin{pmatrix} 0 & \\ & \Sigma_{22}^{-1} \end{pmatrix} \frac{1}{\sqrt{m}} \sum_{t=p+1}^m \mathbf{Z}_t(\boldsymbol{\theta}_0) + o_p(1) \\
&= \tilde{\boldsymbol{W}}(1)^\top \Sigma_{22}^{-1} \tilde{\boldsymbol{W}}(1) + o_p(1) \\
&\xrightarrow{d} \chi_p^2, \text{ as } n \to \infty.
\end{aligned}
$$

This completes the proof of this Theorem. □

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
