# Peer review of "A Unified Test for the AR Error Structure of an Autoregressive Model"

_axioms, doi:10.3390/axioms11120690_

Round 1
Reviewer 1 Report
Dear Authors,
all comments are already mentioned in your manuscript.

Reviewer 2 Report
1. p.2.Please state the definition of e in Equation (2).
2. ln93. "sum least squares," should be "sum of least squares,"
3. ln98, I believe case (ii) is the second line of the above equation but it should be clarified what Case (ii) is.
4. ln211-213. It should explain that since the probability is higher for the right order in Table 4. I think for first time reader it is difficult to interpret from the Table.
5. p.9, Table 5. It would be nice if the authors can provide explanations why the parameters were insignifcant.
6. I hope the authors can include more explanations on how the study results can contribute to the related study field in the conclusion section.
Round 2
Reviewer 3 Report
Congratulations to the authors for the efforts improving their paper.